# Characteristics of High-Level Aminoglycoside-Resistant *Enterococcus faecalis* Isolated from Bulk Tank Milk in Korea

**DOI:** 10.3390/ani11061724

**Published:** 2021-06-09

**Authors:** Hyo Jung Kang, Sunghyun Yoon, Koeun Kim, Young Ju Lee

**Affiliations:** 1College of Veterinary Medicine & Zoonoses Research Institute, Kyungpook National University, Daegu 41566, Korea; sa01083@knu.ac.kr (H.J.K.); goguma0707@gmail.com (S.Y.); kke02062@gmail.com (K.K.); 2Division of Microbiology, National Center for Toxicological Research, U.S. Food and Drug Administration, Jefferson, AR 72079, USA

**Keywords:** high-level aminoglycoside resistance, *Enterococcus faecalis*, bulk tank milk

## Abstract

**Simple Summary:**

Aminoglycosides are used to treat various infections in veterinary and human medicine. However, with the emergence of high-level aminoglycoside resistance in human and food-producing animals, the synergism of aminoglycosides with beta-lactam or glycopeptide is being threatened. Moreover, the environmental mastitis-causing agent, enterococci, has emerged as a cause of nosocomial infection due to its antimicrobial resistance. Therefore, the purpose of this study was to investigate the characteristics of high-level aminoglycoside-resistant *Enterococcus faecalis* isolated from bulk tank milk in Korea. It showed that 185 (61.5%) isolates out of 301 were high-level aminoglycoside resistant, while 149 isolates were multidrug resistant.

**Abstract:**

Enterococci, which are considered environmental mastitis-causing pathogens, have easily acquired aminoglycoside-resistant genes that encode various aminoglycoside-modifying enzymes (AME). Therefore, this study was conducted to compare the distribution of high-level aminoglycoside-resistant (HLAR) and multidrug-resistant (MDR) *Enterococcus faecalis* (*E. faecalis*) bacteria isolated from bulk tank milk in four dairy companies in Korea. Moreover, it analyzed the characteristics of their antimicrobial resistance genes and virulence factors. Among the 301 *E. faecalis* bacteria studied, 185 (61.5%) showed HLAR with no significant differences among the dairy companies. Furthermore, 129 (69.7%) of the 185 HLAR *E. faecalis* showed MDR without significant differences among companies. In contrast, HLAR *E. faecalis* from companies A, B, and C were significantly higher in resistance to the four classes than those in company D, which had the highest MDR ability against the three antimicrobial classes (*p* < 0.05). In addition, in the distribution of AME genes, 72 (38.9%) and 36 (19.5%) of the isolates carried both *aac(6′)Ie-aph(2″)-la* and *ant(6)-Ia* genes, and the *ant (6)-Ia* gene alone, respectively, with significant differences among the companies (*p* < 0.05). In the distribution of virulence genes, the *ace* (99.5%), *efa A* (98.9%), and *cad 1* (98.4%) genes were significantly prevalent (*p* < 0.05). Thus, our results support that an advanced management program by companies is required to minimize the dissemination of antimicrobial resistance and virulence factors.

## 1. Introduction

Aminoglycosides are antimicrobials, including gentamicin, streptomycin, and kanamycin, that inhibit bacterial protein synthesis [1]. In particular, aminoglycosides are used in the treatment of aerobic Gram-negative bacilli infections, and with broad-spectrum beta-lactam for severe infections [2]. However, the emergence of resistance to aminoglycosides has continuously been reported among isolates from humans and food-producing animals. This resistance has also been associated with exposure to the commonly used agents [3,4].

Enterococci have increasingly emerged as a cause of serious nosocomial infection in humans and have also been considered environmental mastitis-causing pathogens in veterinary medicine [5,6]. Although synergic combinations of penicillin or a glycopeptide with an aminoglycoside have been used for treating such infections, enterococci have easily acquired aminoglycoside-resistant genes that encode various aminoglycoside-modifying enzymes (AME). These acquired genes cause high resistance to aminoglycosides [7]. In particular, high-level resistance to aminoglycosides can abolish the synergic effect between commercially available aminoglycosides and cell-wall active agents, such as beta-lactams or glycopeptides [8].

Although high-level aminoglycoside-resistant (HLAR) enterococci were first reported in the 1980s in humans [9], and have been described in several studies investigating antimicrobial resistance profiles in raw milk or dairy products worldwide [10,11], there have been no comprehensive surveys to date on the characteristics of HLAR enterococci obtained from raw milk or dairy products in Korea. Hence, this study was conducted to compare the distribution of HLAR and multidrug-resistant (MDR) *Enterococcus faecalis* (*E. faecalis*) isolated from bulk tank milk in four major dairy companies in Korea. Moreover, it analyzed the characteristics of the antimicrobial resistance genes and virulence factors of the bacterial strains of interest.

## 2. Materials and Methods

### 2.1. Bacterial Isolation

A total of 1584 batches of bulk tank milk samples from 395 farms belonging to four dairy companies in Korea were collected twice during the summer and winter each (July–December 2019). Then, 50 mL of milk samples were aseptically collected from each bulk sample and sent to the laboratory at 4 °C. For the isolation and identification of *E. faecalis*, 1 mL of the milk sample was cultured in 9 mL buffered peptone water (BPW; BD Biosciences, San Jose, CA, USA); the pre-enriched BPW was then mixed with an Enterococcosel broth (BD Biosciences) at a 1:10 ratio, and incubated at 37 °C for 18–24 h. Furthermore, each medium was streaked onto an Enterococcosel agar (BD Biosciences), and confirmation of *E. faecalis* was performed using PCR, as described previously [12]. Among the isolates showing the same antimicrobial susceptibility patterns from the same origin, only one isolate was chosen for this study. As a result, *E. faecalis* isolates were tested.

### 2.2. Antimicrobial Susceptibility Testing

According to the guidelines of the Clinical and Laboratory Standards Institute (CLSI, 2019) [13], the disk diffusion method was performed for all *E. faecalis* isolates against 11 antimicrobial agents (BD Bioscience, Sparks, MD, USA). These 11 antimicrobial agents are: ampicillin (AM, 10 μg), chloramphenicol (C, 30 μg), ciprofloxacin (CIP, 5 μg), doxycycline (DOX, 30 μg), erythromycin (E, 15 μg), high-level gentamicin (G, 120 μg), penicillin (P, 10 units), rifampin (RA, 5 μg), high-level streptomycin (S, 300 μg), tetracycline (TE, 30 μg), and vancomycin (VA, 30 μg). MDR was defined as acquired resistance to at least one agent of the three or more antimicrobial classes [14].

### 2.3. Detection of HLAR Enterococci

The standard agar dilution method conducted on brain heart infusion agar was used to determine the minimum inhibitory concentration values for G and S, with a concentration range of 256–2048 μg/mL (serial 2-fold dilutions). Moreover, breakpoints for high-level G and S were set at ≥500 and ≥2000 μg/mL, respectively, following the CLSI guidelines (CLSI, 2019) [13].

### 2.4. Detection of Antimicrobial Resistance and Virulence Genes

The presence of genes conferring resistance to aminoglycosides (*aac (6″)Ie-aph(2″)-la, ant (6)-Ia, aph(2″)-Ic,* and *aph (2″)-Id*), macrolide (*erm A, erm B,* and *mef*), oxazolidinone (*optr A* and *poxt A*), phenicols (*cat A*, *cat B, cfr,* and *fex A*), and tetracyclines *(tet L, tet M,* and *tet O)* were investigated using PCR, as described previously [15,16,17,18,19,20,21]. Genes encoding virulence factors such as collagen-binding protein (*ace*), aggregation substance (*asa 1*), pheromone cAD1 precursor lipoprotein (*cad 1*), cytolysin (*cyl A activator*), *E. faecalis* endocarditis antigen (*efa A*), enterococcal surface protein (*esp*), and gelatinase (*gel E*) were also detected, as described previously [22,23]. The primers used in this study are presented in Table 1.

### 2.5. Statistical Analysis

The Statistical Package for Social Science version 25 (IBM SPSS Statistics for Windows, Armonk, NY, USA) was used for statistical analysis. Further, Pearson’s chi-square test and Fisher’s exact test with Bonferroni correction were used to compare the prevalence of isolates between companies [25]. A *p* value < 0.05 was considered statistically significant.

## 3. Results

### 3.1. Prevalence of MDR and HLAR E. faecalis

The distribution of MDR and HLAR in *E. faecalis* from the bulk tank milk of four dairy companies is presented in Table 2. Among the 301 *E. faecalis* isolates studied, 149 (49.5%) and 185 (61.5%) showed MDR and HLAR, respectively. Moreover, although company D showed the highest prevalence of *E. faecalis*, the prevalence of MDR *E. faecalis* was significantly higher in isolates from company A (61.5%) (*p* < 0.05). However, the prevalence of HLAR *E. faecalis* showed no significant difference among the four dairy companies.

### 3.2. Antimicrobial Resistance of HLAR E. faecalis

The distribution of resistance against nine antimicrobial agents of 185 HLAR *E. faecalis* is presented in Table 3. The results showed that the significantly highest resistance was against TE (93.5%), followed by E (71.9%), then DOX (70.8%). In particular, resistance to DOX also showed significant differences among the dairy companies (*p* < 0.05). However, resistance to AM, CIP, P, RA, and VA was only 0% to 5.9%.

### 3.3. Distribution of MDR Patterns

The distribution of MDR isolates among 185 HLAR *E. faecalis* is presented in Figure 1. Although the prevalence of MDR (129 isolates, 69.7%) in HLAR *E. faecalis* showed no significant differences among the dairy companies, HLAR *E. faecalis* from company A showed the highest MDR (80.6%), followed by company D (73.2%), C (62.9%), and B (60.7%), respectively. Likewise, all MDR isolates showed resistance against three to five antimicrobial classes. In particular, HLAR *E. faecalis* from companies A, B, and C were significantly higher in resistance against the four classes than company D, which showed the highest MDR against only three of the antimicrobial classes (*p* < 0.05). Furthermore, MDR to five classes was observed only in HLAR *E. faecalis* from companies A (5.6%) and C (3.7%).

### 3.4. Distribution of Antimicrobial Resistance Genes

The distributions of resistance genes in 185 HLAR *E. faecalis* are presented in Table 4. In the distribution of AME genes, 72 (38.9%) and 36 (19.5%) isolates, respectively, expressed both *aac (6′)Ie-aph(2″)-la* and *ant (6)-Ia* genes, and the *ant (6)-Ia* gene alone, with significant differences among the dairy companies (*p* < 0.05). Likewise, for the E resistance genes, the *erm B* gene (71.4%) among the three genes had the highest prevalence (*p* < 0.05), although no significant difference among the dairy companies was observed. Moreover, in the tetracycline resistance genes, the prevalence of both *tet M* and *tet L* genes (46.5%), as well as the *tet M* gene (36.3%) alone, had the highest prevalence with significant differences among the studied dairy companies (*p* < 0.05). Furthermore, the *cat A* and *cfr* genes related to resistance to C were observed among 27 (14.6%) and two (1.1%) isolates, respectively. In contrast, the *optr A* and *poxt A* genes related to resistance to linezolid were observed only in two (1.1%) isolates and one (0.5%) isolate, respectively.

### 3.5. Distribution of Virulence Genes

The distributions of virulence genes in 185 HLAR *E. faecalis* are presented in Table 5. The *ace* (99.5%), *efa A* (98.9%), and *cad 1* (98.4%) genes were the prevalent genes (*p* < 0.05), followed by the *gel E* (85.9%), *asa 1* (61.6%), *esp* (12.4%), and *cyl A* (6.5%) genes. However, significant differences among the dairy companies were observed in *efa A*, *cyl A,* and *gel E* genes (*p* < 0.05).

## 4. Discussion

In Korea, five major dairy companies produce 84% of the total milk and dairy products consumed. Confinement housing is used to run most of the farms, which is a primary management system of dairy production [26]. Washburn et al., (2002) [27] reported that confined cows had 1.8 times more clinical mastitis compared with cows on pasture. Therefore, various antimicrobials have been used for treating mastitis every year in Korea [28]. In particular, aminoglycosides, which are used along with cell-wall active agents, are effective for treating serious enterococcus infection [29]. However, enterococci showing high resistance to aminoglycosides has been reported continuously in food-producing animals [30,31,32]. In this study, 61.5% of the 301 *E. faecalis* isolates from bulk tank milk were HLAR, although no significant difference among the four companies was shown. Chajęcka-Wierzchowska et al., (2020) [10] reported that 30.2% of the ready-to-eat dairy products in Poland showed HLAR to enterococci. Özdemir and Tuncer (2020) [33] also reported that 59 HLAR enterococci were observed in 100 samples of milk and dairy in Turkey. Furthermore, the high prevalence of HLAR enterococci in Korea compared with that in Poland and Turkey is indirect proof of the consistent use of aminoglycosides to treat bacterial infection in Korea.

Interestingly, 69.7% of the HLAR isolates showed MDR in this study. Hegstad et al. (2010) [34] reported that enterococci have a common feature of easily transferring DNA via plasmid or transposons encoding resistance genes. Therefore, the capacity of transference to other bacteria by these plasmids or transposons leads to the spread of various resistance strains, which ultimately results in MDR. The distribution of MDR patterns in HLAR *E. faecalis* showed significant differences among the companies in this study. Likewise, HLAR *E. faecalis* from companies A, B, and C showed the highest prevalence in MDR against the four classes. In contrast, isolates from company D showed the highest prevalence in the three classes. MDR of five classes was observed only in HLAR *E. faecalis* from companies A and C. These results suggest that the critical point for reducing the emergence of resistant bacteria is the management of which and how dairy companies use antimicrobials.

Hollenbeck and Rice (2012) [35] reported that all enterococci possess intrinsic low-level resistance to all aminoglycosides by limiting the uptake of drugs, and this resistance originated from their facultative anaerobic metabolism. However, genes encoding diverse AME acquire high-level resistance to aminoglycosides of enterococci. Thus, it eliminates the synergism of aminoglycosides with cell-wall synthesis interfering agents, such as β-lactams [36,37,38]. In this study, the combination of both *aac (6′)Ie-aph(2″)-la* and *ant (6)-Ia* genes was significantly prevalent (*p* < 0.05). The *ant (6)-Ia* gene is responsible for encoding the ANT (O-adenyltransferase) enzyme that the catalyzes ATP-dependent adenylation of a hydroxyl group, and grant resistance to streptomycin without cross-resistance to other aminoglycosides [39]. In contrast, *aac (6′)Ie-aph**(2″)-la* encodes the bifunctional enzymes AAC (N-Acetyltransferase) and APH (O-Phosphotransferase), which are responsible for resistance to all types of aminoglycosides, except streptomycin and spectinomycin [40]. Therefore, the high prevalence of HLAR in this study can be related to the distribution of these genes.

In this study, HLAR *E. faecalis* showed the highest resistance to TE (93.5%), followed by E (71.9%). The most prevalent of the different types of antimicrobial resistance genes were *tet M* (82.7%), including a combination of *tet M* and *tet L*, which is related to resistance to tetracyclines, and *erm B* (71.4%), which has shown resistance to macrolides. Although the high distribution of these genes has been continuously reported in enterococci isolated from humans and food-producing animals [41,42], it is important that the *erm B* and *tet* (M) genes can also be transferred easily by conjugative transposons, such as the Tn*916/1545* and Tn*5397* families [43]. Therefore, the horizontal transfer of these genes in enterococci should be continuously monitored in the future. In Korea, tetracyclines as feed additives have been banned since 2009, but a large amount of chlortetracycline calcium, chlortetracycline HCL, oxytetracycline dehydrate, and oxytetracycline HCL have still been used for treating mastitis [44]. Furthermore, E is rarely used in the dairy industry [25,44], but resistance to E is linked to the use of tylosin, which is a macrolide, and is used widely for treating streptococcal mastitis [45]. Therefore, our results support that acquiring resistance genes in food-producing animals was induced by the use of antimicrobials, which also contribute to the burden of growing antimicrobial resistance in humans.

Furthermore, due to the acquisition of virulence genes being directly related to the capability of bacteria to cause illness [46], monitoring virulence genes of enterococci is crucial regarding the public health concerns of dairy products. In this study, HLAR *E. faecalis* showed a high prevalence of genes such as *ace* (99.5%), *efa A* (98.9%), *cad 1* (98.4%), *gel E* (85.9%), and *asa 1* (61.6%). This result was in accordance with the high prevalence of virulence genes of *E. faecalis* from buffalo milk in Brazil [47] and *E. faecalis* from dairy products in Egypt [48]. Although the presence of virulence genes does not increase pathogenicity, virulence factors promote tissue colonization in the hosts. Moreover, the combination of antimicrobial resistance genes and virulence factors in enterococci, which are potential opportunistic pathogens regarding clinical or subclinical mastitis, could be a public health problem. Thus, in this study, the phenotypic and genotypic characteristics of HLAR *E. faecalis* showed significant differences among the dairy companies. However, an advanced management protocol by companies is warranted to minimize the dissemination of antimicrobial resistance and virulence factors.

## 5. Conclusions

In this comprehensive research on HLAR, *E. faecalis* isolated from the bulk tank milk of four dairy companies in Korea, the distribution of antimicrobial resistance in these bacteria and the genetic characteristics of HLAR *E. faecalis* showed a significant difference among the companies. Therefore, our results suggest that advanced management programs by companies are warranted to minimize the emergence of antimicrobial-resistant bacteria and to reduce the dissemination of these resistance genes and virulence factors.

## Figures and Tables

**Figure 1 animals-11-01724-f001:**
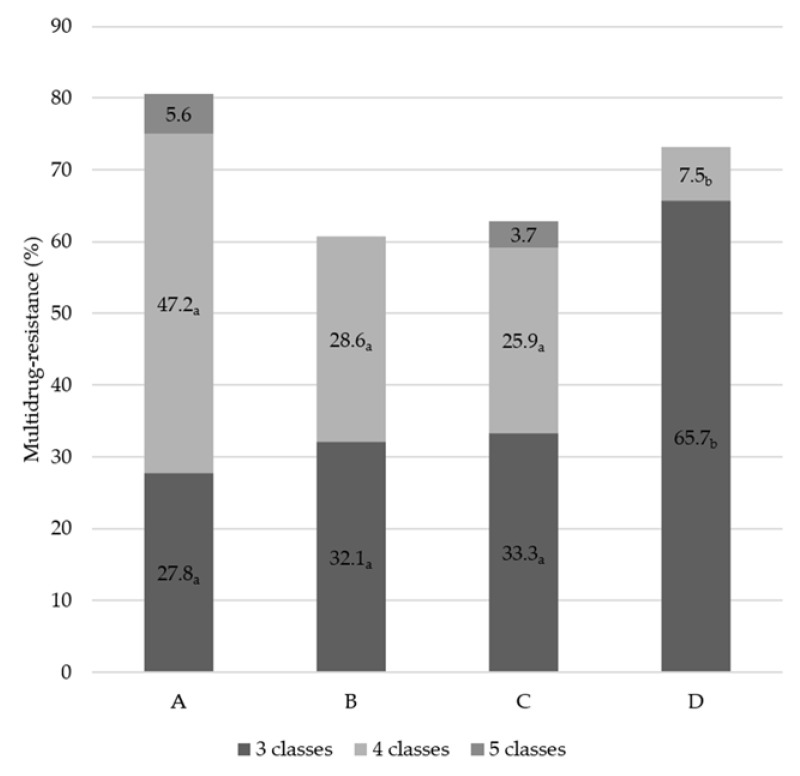
Distribution of multidrug resistance of 185 high-level aminoglycoside-resistant *Enterococcus faecalis* isolated from the bulk tank milk of four dairy companies. Values without the same subscript letter (_a,b_) differ significantly (*p* < 0.05).

**Table 1 animals-11-01724-t001:** Primer sequences used for this study.

Locus	Target Gene	Sequence (5′-3′)	Size (pb)	Reference
Aminoglycoside-modifying enzymes	*aac(6″)Ie-aph(2* *″* *)-1la*	F: CAGAGCCTTGGGAAGATGAAG	348	[17]
		R: CCTCGTGTAATTCATGTTCTGGC		
	*ant(6)-Ia*	F: ACTGGCTTAATCAATTTGGG	597	[15]
		R: GCCTTTCCGCCACCTCACCG		
	*aph(2″)-Ic*	F: CCACAATGATAATGACTCAGTTCCC	444	[17]
		R: CCACAGCTTCCGATAGCAAGAG		
	*aph(2″)-Id*	F: GTGGTTTTTACAGGAATGCCATC	641	[17]
		R: CCCTCTTCATACCAATCCATATAACC		
Macrolide resistance	*ermA*	F: TAACATCAGTACGGATATTG	200	[19]
		R: AGTCTACACTTGGCTTAGG		
	*ermB*	F: CCGAACACTAGGGTTGCTC	139	[19]
		R: ATCTGGAACATCTGTGGTATG		
	*mef*	F: AGTATCATTAATCACTAGTGC	348	[19]
		R: TTCTTCTGGTACTAAAAGTGG		
Oxazolidinone resistance	*optrA*	F: AGGTGGTCAGCGAACTAA	1395	[20]
		R: ATCAACTGTTCCCATTCA		
	*poxtA*	F: TCCACAAAGGATGGGTTATG	1336	[22]
		R: ATGCCCGTATTGGTTATCTC		
Phenicol resistance	*catA*	F: GGATATGAAATTTATCCCTC	486	[21]
		R: CAATCATCTACCCTATGAAT		
	*catB*	F: TGAACACCTGGAACCGCAGAG	482	[21]
		R: GCCATAGTAAACACCGGAGCA		
	*cfr*	F: TGAAGTATAAAGCAGGTTGGGAGTCA	746	[18]
		R: ACCATATAATTGACCACAAGCAGC		
	*fexA*	F: GTACTTGTAGGTGCAATTACGGCTGA	1272	[18]
		R: CGCATCTGAGTAGGACATAGCGTC		
Tetracycline resistance	*tetL*	F: ATAAATTGTTTCGGGTCGGTAAT	1077	[16]
		R: AACCAGCCAACTAATGACAATGAT		
	*tetM*	F: GTTAAATAGTGTTCTTGGAG	657	[16]
		R: CTAAGATATGGCTCTAACAA		
	*tetO*	F: CAATATCACCAGAGCAGGCT	614	[16]
		R: TCC CAC TGT TCC ATA TCG TCA		
Virulence gene	*ace*	F: GGAATGACCGAGAACGATGGC	616	[23]
		R: GCTTGATGTTGGCCTGCTTCCG		
	*asa1*	F: CACGCTATTACGAACTATGA	375	[23]
		R: TAAGAAAGAACATCACCACGA		
	*cad1*	F: TTCCAA AACTACGCACAACA	423	[24]
		R: CTTTTTCAGCAGCATTCACTAATT		
	*cylA*	F: GACTCGGGGATTGATAGGC	688	[23]
		R: GCTGCTAAAGCTGCGCTTAC		
	*efaA*	F: CGTGAGAAAGAAATGGAGGA	499	[23]
		R: CTACTAACACGTCACGAATG		
	*esp*	F: AGATTTCATCTTTGATTCTTG	510	[23]
		R: AATTGATTCTTTAGCATCTGG		
	*gelE*	F: TATGACAATGCTTTTTGGGAT	213	[23]
		R: AGATGCACCCGAAATAATATA		

**Table 2 animals-11-01724-t002:** Distribution of multidrug-resistant and high-level aminoglycoside-resistant *Enterococcus faecalis* isolates from the bulk tank milk of four dairy companies.

Company (No. of Farms)	No. of *E. faecalis*	No. of MDR ^1^ (%)	No. of HLAR ^2^ (%)
A (106)	52	37 (71.2) _a_	36 (69.2)
B (47)	39	20 (51.3) _a,b_	28 (71.8)
C (120)	86	41 (47.7) _b_	54 (62.8)
D (122)	124	51 (41.1) _b_	67 (54.0)
Total (395)	301	149 (49.5)	185 (61.5)

Bulk tank milk samples were collected in summer and winter from each farm. _a,b_ Values in a column without the same subscript letter differ significantly (*p* < 0.05). ^1^ MDR: multidrug resistance. ^2^ HLAR: high-level aminoglycoside resistance.

**Table 3 animals-11-01724-t003:** Antimicrobial resistance of 185 high-level aminoglycoside-resistant *Enterococcus faecalis* isolates from the bulk tank milk of four dairy companies.

	No. (%) of Antimicrobial-Resistant HLAR *E. faecalis* by Company				
Antimicrobials	A (*n* = 36) *	B (*n* = 28)	C (*n* = 54)	D (*n* = 67)	Total (*n* = 185)
Ampicillin	0 (0.0)	1 (3.6)	0 (0.0)	1 (1.5)	2 (1.1) ^A,B^
Chloramphenicol	27 (75.0) _a_	14 (50.0) _a,b_	19 (35.2) _b,c_	15 (22.4) _c_	75 (40.5) ^C^
Ciprofloxacin	0 (0.0)	0 (0.0)	2 (3.7)	0 (0.0)	2 (1.1) ^A,B^
Doxycycline	24 (66.7) _a,b_	20 (71.4) _a,b_	31 (57.4) _b_	56 (83.6) _a_	131(70.8) ^D^
Erythromycin	30 (83.3)	18 (64.3)	43 (79.6)	42 (62.7)	133 (71.9) ^D^
Penicillin	0 (0.0)	0 (0.0)	0 (0.0)	0 (0.0)	0 (0.0) ^B^
Rifampin	4 (11.1)	0 (0.0)	5 (9.3)	2 (3.0)	11 (5.9) ^A^
Tetracycline	33 (91.7)	25 (89.3)	50 (92.6)	65 (97.0)	173 (93.5) ^E^
Vancomycin	0 (0.0)	1 (3.6)	0 (0.0)	0 (0.0)	1 (0.5) ^A,B^

* *n* = No. of high-level aminoglycoside-resistant *Enterococcus faecalis* isolated from bulk tank milk by company. Values with different subscript letters (_a–c_) represent significant differences among farms, while superscript letters (^A–E^) represent the total significant difference (*p* < 0.05).

**Table 4 animals-11-01724-t004:** Distribution of antimicrobial resistance genes of 185 high-level aminoglycoside-resistant *Enterococcus faecalis* from the bulk tank milk of four dairy companies.

Genes	No. (%) of Isolates with Antimicrobial Resistance Gene(s) by Company				
	A (*n* = 36) *	B (*n* = 28)	C (*n* = 54)	D (*n* = 67)	Total (*n* = 185)
*Aminoglycoside-modifying enzymes*					
*aac(6′)Ie-aph(2″)-la*	2 (5.6)	1 (3.6)	6 (11.1)	4 (6.0)	13 (7.0) ^A^
*ant(6)-Ia*	8 (22.2) _a,b_	10 (35.7) _b_	13 (24.1) _a,b_	5 (7.5) _a_	36 (19.5) ^B^
*aph(2″)-Ic*	0 (0.0)	1 (3.6)	1 (1.9)	0 (0.0)	2 (1.1) ^A,C^
*aph(2″)-Id*	1 (2.8)	2 (7.1)	2 (3.7)	8 (11.9)	13 (7.0) ^A^
*aac(6″)Ie-aph(2″)-la, ant(6)-Ia*	12 (33.3) _a,b_	6 (21.4) _b_	18 (33.3) _a,b_	36 (53.7) _a_	72 (38.9) ^D^
*aac(6″)Ie-aph(2″)-la, aph(2″)-Id*	0 (0.0)	0 (0.0)	1 (1.9)	0 (0.0)	1 (0.5) ^C^
*aph(2″)-Ic, aph(2″)-Id*	0 (0.0)	0 (0.0)	1 (1.9)	0 (0.0)	1 (0.5) ^C^
*aph(2″)-Ic, ant(6)-Ia*	1 (2.8)	0 (0.0)	1 (1.9)	1 (1.5)	3 (1.6) ^A,C^
*aph(2″)-Id, ant(6)-Ia*	1 (2.8)	3 (10.7)	5 (9.3)	3 (4.5)	12 (6.5) ^A,C^
*aph(2″)-Ic, aph(2* *″)-Id, ant(6)-Ia*	0 (0.0)	0 (0.0)	1 (1.9)	0 (0.0)	1 (0.5) ^C^
Macrolides					
*ermA*	0 (0.0)	1 (3.6)	0 (0.0)	0 (0.0)	1 (0.5) ^A^
*ermB*	29 (80.6)	18 (64.3)	43 (79.6)	42 (62.7)	132 (71.4) ^B^
*mef*	0 (0.0)	0 (0.0)	0 (0.0)	0 (0.0)	0 (0.0) ^A^
Oxazolidinones					
*optrA*	0 (0.0)	1 (3.6)	0 (0.0)	1 (1.5)	2 (1.1) ^A^
*poxtA*	0 (0.0)	0 (0.0)	1 (1.9)	0 (0.0)	1 (0.5) ^A^
Phenicols					
*catA*	11 (30.6) _a_	3 (10.7) _a,b_	8 (14.8) _a,b_	5 (7.5) _b_	27 (14.6) ^B^
*catB*	0 (0.0)	0 (0.0)	0 (0.0)	0 (0.0)	0 (0.0) ^A^
*cfr*	2 (5.6)	0 (0.0)	0 (0.0)	0 (0.0)	2 (1.1) ^A^
*fexA*	0 (0.0)	0 (0.0)	0 (0.0)	0 (0.0)	0 (0.0) ^A^
Tetracyclines					
*tetL*	0 (0.0)	1 (3.6)	3 (5.6)	2 (3.0)	6 (3.2) ^A^
*tetM*	19 (52.8) _a_	3 (10.7) _b_	30 (55.6) _a_	15 (22.4) _b_	67 (36.2) ^B^
*tetO*	0 (0.0)	2 (7.1)	0 (0.0)	0 (0.0)	2 (1.1) ^A^
*tetM + tetL*	11 (30.6) _a_	15 (53.6) _a,b_	16 (29.6) _a_	44 (65.7) _b_	86 (46.5) ^B^

* *n* = No. of high-level aminoglycoside-resistant *Enterococcus faecalis* isolated from bulk tank milk by company. Values with different subscript letters (_a,b_) represent significant differences among farms, while superscript letters (^A–D^) represent the total significant difference (*p* < 0.05).

**Table 5 animals-11-01724-t005:** Distribution of virulence genes of 185 high-level aminoglycoside-resistant *Enterococcus faecalis* isolated from the bulk tank milk of four dairy companies.

Genes	No. (%) of Isolates with Virulence Gene(s) by Company				
	A (*n* = 36)	B (*n* = 28)	C (*n* = 54)	D (*n* = 67)	Total (*n* = 185)
*ace*	36 (100)	27 (96.4)	54 (100)	67 (100)	184 (99.5) ^A^
*asa1*	24 (66.7)	17 (60.7)	28 (51.9)	45 (67.2)	114 (61.6) ^B^
*cad1*	36 (100)	28 (100)	53 (98.1)	65 (97.0)	182 (98.4) ^A^
*cylA*	5 (13.9) _a_	2 (7.1) _a,_b	5 (9.3) _a,b_	0 (0) _b_	12 (6.5) ^C^
*efaA*	36 (100) _a_	26 (92.9) _b_	54 (100) _a_	67 (100) _a_	183 (98.9) ^A^
*esp*	4 (11.1)	4 (14.3)	11 (20.4)	4 (6.0)	23 (12.4) ^C^
*gelE*	31 (86.1) _a,b_	23 (82.1) _a,b_	40 (74.1) _b_	65 (97.0) _a_	159 (85.9) ^D^

Values with different subscript letters (_a,b_) represent significant differences among farms, while superscript letters (^A–D^) represent the total significant difference (*p* < 0.05).

## Data Availability

Not applicable.

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
