# Peer review of "Characteristics of High-Level Aminoglycoside-Resistant *Enterococcus faecalis* Isolated from Bulk Tank Milk in Korea"

_animals, 2021, doi:10.3390/ani11061724_

Round 1
Reviewer 1 Report
see attached report

Author Response
Thank you for your detailed review of my article.

Reviewer 2 Report
Abstract
The abstract should start by mentioning clearly the objectives of the study (see below).
Introduction.
The first paragraph contains very basic knowledge and is really redundant. It should be deleted.
The text needs to start from the sentences: Enterococci have increasingly emerged as a cause of serious ….
The current European Union official documentation should also be referred to. Please remember that this is an international journal and the EU is the only international entity that has both scientific AND legislative documentATION about resistance to antimicrobials.
The objectives of the study should be clearly mentioned as follows: The objectives of the study were ……..
Materials and methods
Please list the criteria for selection, inclusion and exclusion, of the 395 farms.
Please include a map with all the locations of the farms.
Table 1. Please move as supplementary material.
Results
Please provide 95% confidence intervals for all proportions listed.
Figure 1. Please colourise rather than using shades of gray.
Please provide exact P values, a dichotomous approach is not acceptable at all, as it raises issues. If this issue is corrected, it will lead directly to rejection at the next round of evaluation.
Please provide an analysis by geographical location of the farm.
Analysis by dairy companies (3.1.) did not offer any valuable results. It should be deleted or at worst transferred in supplementary material. Also, I cannot see the value for the DOX difference among companies; it is said P<0.05, how much exactly was this significance? Really this analysis is totally wrong and does not offer anything useful.
Associations should be presented between phenotyping findings, AMR genes and virulence genes.
Discussion
The discussion should be written from scratch based on the new analysis and results as indicated above.
The manuscript is important, but the authors have taken this entirely wrong approach with the five dairy companies and decreased the value. This approach is superficial and erroneous. The authors should change their approach. An entirely new statistical analysis must be made and presented. Associations should be evaluated.
As it is now, the manuscript appears to be of little interest.
The authors must make the changes and resubmit a correctly revised manuscript for further evaluation. I foresee another 2 to 3 more rounds of evaluation before possible acceptance.
Author Response

(The authors gave the same response as above.)

Round 2
Reviewer 2 Report
This is a fantastic work that clearly should be published as top priority in the journal.
An excellent study with fantastic methodologies, that have been used very well in this context.
The authors should be commended for performing such top-class work.
The work has been well-planned, well-executed and well-presented.
This manuscript can be published as it is.
Congratulations